# Acneiform Eruptions Caused by Lithium Treatment May Be Related to *Demodex* Mites (Prostigmata: Demodecidae): A Hypothesis

**DOI:** 10.3390/biology14060605

**Published:** 2025-05-25

**Authors:** Rif S. El-Mallakh, Masoud Doroodgar, Mehak Pahwa, Omar H. A. Elsayed

**Affiliations:** Mood Disorders Research Program, Depression Center, Department of Psychiatry and Behavioral Sciences, University of Louisville School of Medicine, Louisville, KY 40202, USA; masouddoroodgar@yahoo.com (M.D.); mehak.pahwa@louisville.edu (M.P.); omarheshamabdelkhalek.elsayed@louisville.edu (O.H.A.E.)

**Keywords:** acne, acne vulgaris, *Demodex*, lithium, sweat

## Abstract

Lithium is a treatment for people with bipolar disorder but may be associated with onset acne vulgaris. It is proposed that this adverse consequence of lithium treatment may be related to an interaction between lithium and *Demodex* mites. It is proposed that the hypothesis can be tested by using acaricidal agents to treat lithium-related acneiform eruptions.

## 1. Introduction

Acne vulgaris is a common and burdensome skin disease that involves the pilosebaceous unit with altered keratinization, inflammation, and bacterial colonization of hair follicles, usually by *Propionibacterium acnes*. The disease manifests with maculopapular eruptions of the face, neck, chest, and back where the sebaceous glands are large and numerous [1]. It afflicts some 20% of teenagers but can persist into adulthood [1]. The pathoetiology of acne vulgaris is not known but believed to involve altered sebum production possibly fueled by increased sex hormone production in early adulthood [1].

*Demodex* mites (Prostigmata: Demodecidae) are human ectoparasites that are present in most humans [2]. There are two species of human-colonizing *Demodex* that have been identified, *Demodex folliculorum*, which lives predominantly in hair follicles, and *Demodex brevis*, which lives in sebaceous glands. The majority of affected people do not display any symptoms, i.e., the presence of *Demodex* is commensal [3], but on occasion *Demodex* has been associated with some skin conditions, including acne [4,5]. It has been proposed that *D. folliculorum* is evolutionarily transitioning from a host-injuring obligate parasite to an obligate symbiont [2].

Lithium is the lightest alkali metal and has documented utility in the management of the psychiatric illness of manic-depressive disorder or bipolar disorder [6]. Acneiform eruptions are a common adverse effect in patients receiving lithium for bipolar illness [7], with a prevalence of 33% in lithium-treated patients compared to 9% in matched non-lithium-treated mood-disordered patients (*p* = 0.0016) [8]. It is not clear why lithium is associated with acne, but its occurrence is associated with significant morbidity and it often requires discontinuation of the medication [7]. In this paper, we propose that lithium-related acne may be a consequence of the effects of lithium on *Demodex*.

## 2. Materials and Methods

This was a directed review and not a comprehensive or a systematic review. Due to the proliferation of predatory journals, we limited our searches to PubMed because the National Library of Medicine attempts to ensure that only adequately peer-reviewed journals are indexed there. We searched the terms (Demodex and sweat = 3), (Demodex and lithium = 0), (Demodex and acne = 84), (lithium and acne = 84), and (sweat and acne = 187).

## 3. Results and Discussion

The majority of published work regarding the relationship between *Demodex* and pathologic presentations has dealt with blepharitis. Blepharitis is inflammation of the eyelid, and it presents with erythema, swelling, pain, irritation, and possibly discharge of the eyelid margin and lashes, and other eyelash abnormalities. A common cause of blepharitis, accounting for as much as 60% of clinical cases, is *Demodex* infestation [9]. Caution needs to be exercised because the presence of *Demodex* is nearly universal, particularly in older individuals, so that they are present in people with and without blepharitis. However, some authors argue that while lower numbers of *Demodex* mites may be benign or even beneficial, very high numbers are associated with pathology [9]. A number of ≥5 *Demodex* individuals per cm^2^ on a standardized skin surface biopsy (SSSB) is frequently considered pathologic [10]. (The SSSB is performed by placing a glass slide with 1 cm^2^ coated with cyanoacrylate glue on the skin for 1 min and then examining it under a microscope.) Such an expanded number of mites is called demodicosis. *Demodex* cause problems by inducing inflammation, and they may also act as fomites for *Streptococcus* and *Staphylococcus* bacteria, which may play a role in blepharitis [11]. Because of the potential beneficial role of mites, the treatment of blepharitis is recommended to be a knockdown of the number of mites in the skin rather than eradication of the commensal creature [12].

The two species of Demodex cause different conditions due to their habits and biology. *D. folliculorum*, which causes anterior blepharitis, produces inflammation in the root of the eye lash. This is related to the waste products of mite feeding (since *Demodex* do not have anuses, they regurgitate undigested material) that may combine with uneaten parts of epithelial cell debris and other mite material such as eggs to form a deposit at the base of the eyelash called cylindrical dandruff, with consequent irritation [11]. *D. brevis* is more likely to cause meibomian gland dysfunction because the mite can physically block drainage of the gland, which leads to swelling and inflammation [13]. Both living and dead mites have been associated with pathological states, with the physical breakdown of dead mites associated with inflammation in response to bacteria that break down the mites [14]. These same processes are believed to lead to skin conditions such as acne vulgaris [15,16,17,18] and rosacea [5,17,19] and nonspecific facial dermatitis [20].

*Demodex* mite populations have been found to be associated with multiple skin conditions [4,5]. The association appears strongest with acne vulgaris [15,16,17,18] and rosacea [5,17,19]. For acne vulgaris, the association was significant but there was no test of principle by the use of permethrin. In a recent case-control study with 40 patients with acne vulgaris, 40 patients with nonspecific facial dermatitis, and 40 patients without skin issues, high *Demodex* mite density was more likely in patients with nonspecific facial dermatitis versus patients with acne vulgaris or normal controls [20]. In a meta-analysis that encompassed 42,130 individuals, the pooled odds ratio (OR) of a significant association in random effects models was 2.80 (95% confidence interval [CI], 2.34–3.36, *p* < 0.05) [21]. To examine the effect of publication bias, the authors calculated a fail-safe number of 18,477, meaning that at least that many articles with negative conclusions would be needed to discredit the current relationship [20]. When treatment with antibiotic, topical metronidazole, was compared with an insecticide, topical permethrin, the outcome was equivalent [22,23].

*Demodex* infestations are common (23–100%) but usually asymptomatic [24]. Moreover, symptomatic demodicosis has been reported to mimic a large number of dermatologic conditions [25,26,27]. Consequently, studies that simply document associations are inadequate, and proof of concept must be obtained from the improvement of the lithium-associated acneiform eruptions responding to acaricidal treatment. That has not yet been examined.

Sodium excretion is important for the formation of sweat, and lithium is known to move through the sodium channels [28], and the concentration of lithium in human sweat may be some fourfold greater than serum levels in these patients [29]. Since lithium has greater energy of hydration than sodium [30], individuals will sweat a greater amount, which could alter the environment for the mites, thereby increasing risk for dermal infection presenting as acneiform eruptions (Figure 1). Lithium is known to increase neutrophil production [31], and this effect has been associated with protection from certain infections [32,33]. It has been proposed that in the pathophysiology of acne that lithium may potentiate conditions that are characterized by neutrophilic infiltration [8].

It is proposed that lithium treatment may interact with the biology of *Demodex* mites. For example, lithium may potentiate *Demodex* survival and lead to increased mite populations with an increased risk of acneiform eruptions or nonspecific dermatitis (Figure 1). If that is the case, a topical insecticide, such as permethrin, would be expected to improve acneiform eruptions associated with lithium. Studies comparing permethrin with metronidazole found that the outcomes were equivalent [22,23], suggesting that reducing *Demodex* populations may have a positive impact on acneiform eruption.

Recently, lithium was found to kill the honeybee hemolymph-sucking parasitic mite, *Varroa destructor*, when it was fed or trickled to bees [34,35]. This suggests a significant sensitivity to lithium toxicity in these parasitic mites, which are not related to *Demodex*. The level achieved in lithium-treated bees (after 4 days of feeding with syrup spiked with 25 mM of lithium chloride) was 130 mg/kg (control bees had 0.15 mg/kg [36]). Humans receiving lithium for bipolar disorder may have a serum level of lithium that spans 0.6–1.2 mM. Assuming a human level of 1.0 mM in the plasma, and an 87 kg person with 62 L volume, the equivalent human concentration is 0.0042 mg/kg, a much lower level, but humans are also less dense than bees (1.3 g/mL versus 4.76 g/mL). It is possible that lithium may impact *Demodex* survival in a biphasic fashion, improving survival at low concentrations, perhaps due to changes in sweat composition, and reducing survival at higher concentrations due to a direct toxic effect that happens in *Varroa* mites infesting bees.

## 4. Conclusions

Independent of potential mechanisms, a significant expansion in the population of the *Demodex* mites in any particular human individual, appears to be associated with converting a commensal relationship into a pathologic one. The direct presence of a larger population of living mites or associated bacteria, an increase in the waste product of mites, or an immunologic response to the bacteria, could contribute to skin disease, which may take the form of inflammation or infection. Lithium may alter the ecology of the follicle or sweat gland to a sufficient degree to result in an expansion of mite populations with consequent skin pathology. This hypothesis can be tested directly, but the authors of this paper do not have the resources to perform such a study.

We propose that since lithium-related acneiform eruptions are expected to be related to physiological alterations induced by lithium, that cleansing treatments, which can reduce the number of *Demodex* mites, would be an inadequate treatment [37]. Thus, these measures could serve as the control arm of a study of an acaricidal treatment, such as permethrin or ivermectin [38]. If *Demodex* are important in the pathoetiology of acne in lithium-treated patients, acaricidal agents should be more effective than cleansing and equal or more effective than traditional treatments (such as antibiotics). Comparative studies of these two approaches are warranted in lithium-associated acneiform eruptions.

## Figures and Tables

**Figure 1 biology-14-00605-f001:**
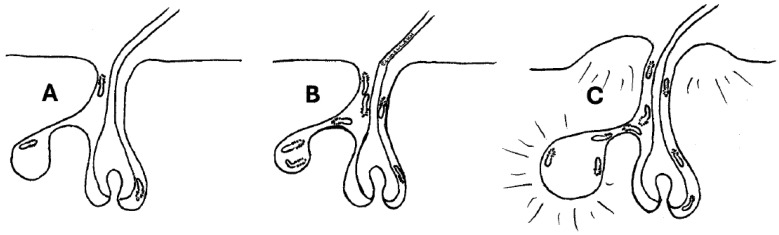
The proposed sequence of events. (**A**) represents the normal commensal hair follicle and associated sebaceous gland, and *Demodex* are present but without pathologic consequences. (**B**) represents an increase in *Demodex* population that may occur due to changes induced in sebum. (**C**) represents the high number of *Demodex* mites and associated waste and possible obstruction of sebum, which may result in inflammation and secondary infection and produce acneiform eruptions.

## Data Availability

Data sharing is not applicable to this hypothesis paper.

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
