# Peer review of "Acneiform Eruptions Caused by Lithium Treatment May Be Related to Demodex Mites (Prostigmata: Demodecidae): A Hypothesis"

_biology, 2025, doi:10.3390/biology14060605_

Round 1

Reviewer 1 Report

Comments and Suggestions for Authors

This manuscript is devoted to acneiform eruptions caused by lithium treatment. The authors hypothesize that it may be related to Demodex mites. Especially authors think that lithium treatment may alter the environment of Demodex mites in a fashion that may increase their propensity to cause disease. Thus treatment with acaricidal agents may be a preferred approach. I think that this manuscript may be published after major revision.

Notes:

1. This manuscript looks like an Opinion but not communication. The authors did not conduct any of their own research here. The article only provides data from other sources. Therefore, this is more like an Opinion.

2. For the better understanding this manuscript should be complemented by the drawings, perhaps from the literature of the references.

3. Authors reported that lithium treatment may interact with the biology of Demodex mites. Is there any concrete evidence of this interaction? References should be provided.

4. At the end of conclusion authors report that this hypothesis can be tested directly. In summing up this study, the authors should either refute their hypothesis, which they put forward at the beginning of the article, or confirm it. The phrase "This hypothesis can be tested directly " is presented too vaguely.

Author Response

         This manuscript is devoted to acneiform eruptions caused by lithium treatment.  The authors hypothesize that it may be related to Demodex mites. Especially authors think that lithium treatment may alter the environment of Demodex mites in a fashion that may increase their propensity to cause disease.  Thus treatment with acaricidal agents may be a preferred approach. I think that this manuscript may be published after major revision.

Notes:

  1. This manuscript looks like an Opinion but not communication. The authors did not conduct any of their own research here. The article only provides data from other sources. Therefore, this is more like an Opinion.

         Response:  The article is a hypothesis presentation.  It reviews the literature to determine if the hypothesis is supported and presents ways to test it.  At any rate, not sure how to address this.

  1. For the better understanding this manuscript should be complemented by the drawings, perhaps from the literature of the references.

         Response:  Figure 1 was added with the following caption:  “Figure 1. The proposed sequence of events.  A. represents the normal commensal hair follicle and associated sebaceous gland.  Demodex are present but without pathologic consequences. B. represents increase in Demodex population that may occur due to changes induced in sebum.  C. The high number of Demodex mites and associated waste and possible obstruction of sebum which may result in inflammation and secondary infection and produce acneiform eruptions.”

  1. Authors reported that lithium treatment may interact with the biology of Demodex mites. Is there any concrete evidence of this interaction? References should be provided.

         Response:  No, there is no evidence that lithium interacts with the mites directly, just that lithium alters the sweat.  We added the following at lines 109-112:  “Since lithium has a larger energy of hydration than sodium[29new], individuals will sweat a greater amount, which could alter the environment for the mites thereby increasing risk for dermal infection presenting as acne (Figure 1).”

  1. At the end of conclusion authors report that this hypothesis can be tested directly. In summing up this study, the authors should either refute their hypothesis, which they put forward at the beginning of the article, or confirm it. The phrase "This hypothesis can be tested directly " is presented too vaguely.

         Response:  We added the following to clarify that the study has not been done (lines 140-141):  “This hypothesis can be tested directly, but the authors of this paper do not have the resources to perform such a study.”

Reviewer 2 Report

Comments and Suggestions for Authors

Dear Authors,

I would like to thank you all to present an important issue. However, I believe, introduction needs to include clearly the aims of the hypothesis in a logical way. In the materials and method section, the systemic review process needs to describe more such as how did you search the literature  or use any tools to find literature. The English language needs to be improved. Authors have used  mixed active and passive sentences in the manuscript. To maintain flow, they should strict one approach.

Comments on the Quality of English Language

The English language needs to be improved. Authors have used  mixed active and passive sentences in the manuscript. To maintain flow, they should strict one approach.

Author Response

I would like to thank you all to present an important issue.

  1. However, I believe, introduction needs to include clearly the aims of the hypothesis in a logical way.

  1. In the materials and method section, the systemic review process needs to describe more such as how did you search the literature or use any tools to find literature.

         Response:  We expanded the Methods section in lines 54-58:  “Due to the proliferation of predatory journals, we limited our searches o PubMed because National Library of Medicine attempts to ensure that only adequately peer-reviewed journals are indexed there.  We searched he terms (Demodex and sweat = 3), (Demodex and lithium =0), (Demodex and acne = 84), (lithium and acne = 84), and (sweat and acne = 187).”

  1. The English language needs to be improved. Authors have used mixed active and passive sentences in the manuscript. To maintain flow, they should strict one approach.

         Response:  We reviewed the paper again, and attempted to improve the language.  We attempted to remove active sentences.

Reviewer 3 Report

Comments and Suggestions for Authors

This is an interesting subject and an interesting hypothesis.

I have some minor comments:

Acneiform lesions or papulopustules that are caused by demodex are referred to as demodicosis, and are not per se acne. Please make sure you refer to those as acneiform eruptions and not acne throughout the paper.

Please see the following paper 'The Association Between Acne Vulgaris, Acne Vulgaris with Nonspecific Facial Dermatitis, and Demodex Mite Presence' from 2024. DOI https://doi.org/10.2147/CCID.S450540 which concludes that demodex mite infestation is not associated with acne.

The metaanalysis you refer to on the same subject is from 2012. I suggest a more specific search of the literature. 

Ivermectin as a topical agent for demodicosis already exists and is an approved treatment. Also, claiming that lithium may lead to dead mites causing a pile up of mite products being treated with a topical that kills mites is somewhat contradictory. Would dead mites respond to agents that kill mites? How will that change the accumulation of mite poducts?

You also do not mention one of the important measures for eradicating demodex mite numbers, including their byproducts - washing the skin instead of using a cotton pad with lotion. Washing with a skin friendly foaming agent would remove dead mites and products, while using permethrin on dead mites would probably not.

I suggest that authors include more details as mentioned above (sweat glands, localization, mite and mite products reducing measures, treatment options for bipolar disease if the condition is severe, etc).

Author Response

This is an interesting subject and an interesting hypothesis.

I have some minor comments:

  1. Acneiform lesions or papulopustules that are caused by demodex are referred to as demodicosis, and are not per se acne. Please make sure you refer to those as acneiform eruptions and not acne throughout the paper.

         Response:  When appropriate we replaced acne with acneiform eruptions.  The exceptions were when we were citing articles in which they studied acne.

  1. Please see the following paper 'The Association Between Acne Vulgaris, Acne Vulgaris with Nonspecific Facial Dermatitis, and Demodex Mite Presence' from 2024. DOI https://doi.org/10.2147/CCID.S450540 which concludes that demodex mite infestation is not associated with acne.

         Response:  This is an important article and was added att lines 90-94:  “A recent case control study with 40 patients with acne vulgaris, 40 patients with nonspecific facial dermatitis and 40 patients without skin issues, high Demodex mite density was more likely in nonspecific facial dermatitis versus acne vulgaris or normal controls [20]”

  1. The metaanalysis you refer to on the same subject is from 2012. I suggest a more specific search of the literature. 

         Response:  That meta-analysis included over 42,000 patients.  The fact that it is 13 years old does not reduce the power of the evaluation, and nothing equivalent has been done since.  No changes were made.

  1. Ivermectin as a topical agent for demodicosis already exists and is an approved treatment.

         Response:  Ivermectin was added as a potential treatment.

  1. Also, claiming that lithium may lead to dead mites causing a pile up of mite products being treated with a topical that kills mites is somewhat contradictory. Would dead mites respond to agents that kill mites? How will that change the accumulation of mite poducts?

         Response:  We clarified that our hypothesis is an expansion of mite population with lithium.

  1. You also do not mention one of the important measures for eradicating demodex mite numbers, including their byproducts - washing the skin instead of using a cotton pad with lotion. Washing with a skin friendly foaming agent would remove dead mites and products, while using permethrin on dead mites would probably not.

         Response:  This is an important point, we added it to the final paragraph in the paper;  “We propose that since lithium-related acneiform eruptions are expected to be related to physiological alterations induced by lithium, that cleansing treatments, which can reduce the number of Demodex mites, would be an inadequate treatment [36].  Thus, these measures could serve as the control arm of a study of an acaricidal treatment, such as permethrin or ivermectin [37].  If Demodex are important in the pathoetiology of acne in lithium-treated patients, acaricidal agents should be more effective than cleansing and equal or more effective than traditional treatments (such as antibiotics).  Comparative studies of these two approaches are warranted in lithium-associated acneiform eruptions.”

  1. I suggest that authors include more details as mentioned above (sweat glands, localization, mite and mite products reducing measures, treatment options for bipolar disease if the condition is severe, etc).

         Response:  We did not add this since this is not a paper about treatment of lithium associated acneiform eruptions but a hypothesis presentation about how that condition might come about.

Round 2

Reviewer 1 Report

Comments and Suggestions for Authors

In this paper, the authors put forward the hypothesis that acneiform eruptions caused by lithium treatment may be related to Demodex mites. Especially authors think that lithium treatment may alter the environment of Demodex mites in a fashion that may increase their propensity to cause disease. Thus treatment with acaricidal agents may be a preferred approach.

Authors have corrected all comments in the paper. I hope these corrections will improve this Communication and the revised version corresponds to high standards of Biology. After careful consideration, I think that this manuscript may be published in this view.